# Alpha Lipoic Acid as a Protective Mediator for Regulating the Defensive Responses of Wheat Plants against Sodic Alkaline Stress: Physiological, Biochemical and Molecular Aspects

**DOI:** 10.3390/plants11060787

**Published:** 2022-03-16

**Authors:** Khaled M. A. Ramadan, Maha Mohammed Alharbi, Asma Massad Alenzi, Hossam S. El-Beltagi, Doaa Bahaa Eldin Darwish, Mohammed I. Aldaej, Tarek A. Shalaby, Abdallah Tageldein Mansour, Yasser Abd El-Gawad El-Gabry, Mohamed F. M. Ibrahim

**Affiliations:** 1Central Laboratories, Department of Chemistry, King Faisal University, Al-Ahsa 31982, Saudi Arabia; kramadan@kfu.edu.sa; 2Biochemistry Department, Faculty of Agriculture, Ain Shams University, Cairo 11566, Egypt; 3Biology Department, Faculty of Science, University of Tabuk, Tabuk 71491, Saudi Arabia; Mmalharbi@ut.edu.sa (M.M.A.); amalanazi@ut.edu.sa (A.M.A.); or d_darwish@mans.edu.eg (D.B.E.D.); 4Agricultural Biotechnology Department, College of Agriculture and Food Sciences, King Faisal University, Al-Ahsa 31982, Saudi Arabia; maldaej@kfu.edu.sa; 5Biochemistry Department, Faculty of Agriculture, Cairo University, Gamma St, Giza 12613, Egypt; 6Botany Department, Faculty of Science, Mansoura University, Mansoura 35511, Egypt; 7Department of Arid Land Agriculture, College of Agricultural and Food Science, King Faisal University, P.O. Box 400, Al-Ahsa 31982, Saudi Arabia; tshalaby@kfu.edu.sa; 8Horticulture Department, Faculty of Agriculture, Kafrelsheikh University, Kafr El-Sheikh 33516, Egypt; 9Animal and Fish Production Department, College of Agricultural and Food Sciences, King Faisal University, P.O. Box 420, Al-Ahsa 31982, Saudi Arabia; amansour@kfu.edu.sa; 10Fish and Animal Production Department, Faculty of Agriculture (Saba Basha), Alexandria University, Alexandria 21531, Egypt; 11Department of Agronomy, Faculty of Agriculture, Ain Shams University, Cairo 11566, Egypt; elgabry@agr.asu.edu.eg; 12Department of Agricultural Botany, Faculty of Agriculture, Ain Shams University, Cairo 11566, Egypt

**Keywords:** *Triticum aestivum* L., high-pH, oxidative damages, ionic homeostasis, osmolytes and qRT-PCR

## Abstract

Recently, exogenous α-Lipoic acid (ALA) has been suggested to improve the tolerance of plants to a wide array of abiotic stresses. However, there is currently no definitive data on the role of ALA in wheat plants exposed to sodic alkaline stress. Therefore, this study was designed to evaluate the effects of foliar application by ALA at 0 (distilled water as control) and 20 µM on wheat seedlings grown under sodic alkaline stress (50 mM 1:1 NaHCO_3_ & Na_2_CO_3_; pH 9.7. Under sodic alkaline stress, exogenous ALA significantly (*p* ≤ 0.05) improved growth (shoot fresh and dry weight), chlorophyll (Chl) a, b and Chl a + b, while Chl a/b ratio was not affected. Moreover, leaf relative water content (RWC), total soluble sugars, carotenoids, total soluble phenols, ascorbic acid, K and Ca were significantly increased in the ALA-treated plants compared to the ALA-untreated plants. This improvement was concomitant with reducing the rate of lipid peroxidation (malondialdehyde, MDA) and H_2_O_2_. Superoxide dismutase (SOD) and ascorbate peroxidase (APX) demonstrated greater activity in the ALA-treated plants compared to the non-treated ones. Conversely, proline, catalase (CAT), guaiacol peroxidase (G-POX), Na and Na/K ratio were significantly decreased in the ALA-treated plants. Under sodic alkaline stress, the relative expression of photosystem II (D2 protein; PsbD) was significantly up-regulated in the ALA treatment (67% increase over the ALA-untreated plants); while Δ pyrroline-5-carboxylate synthase (P5CS), plasma membrane Na^+^/H^+^ antiporter protein of salt overly sensitive gene (SOS1) and tonoplast-localized Na^+^/H^+^ antiporter protein (NHX1) were down-regulated by 21, 37 and 53%, respectively, lower than the ALA-untreated plants. These results reveal that ALA may be involved in several possible mechanisms of alkalinity tolerance in wheat plants.

## 1. Introduction

Wheat (*Triticum aestivum* L.) is one of the three most extensively consumed cereals in the world. It is a major cereal crop used for human nourishment and animal feed all over the world. It is grown in a variety of nations and has a significant impact on the worldwide agricultural economy [1]. Wheat demand in developing countries is predicted to increase by 60% by 2050 due to expected population growth [2]. The gap between the production and consumption of wheat must be filled in order to address future food shortages [3]. Wheat, as well as other cereals and a variety of other crops, is constrained by a variety of abiotic and biotic factors [4].

Soil degradation is accelerated by salinization and sodification, which have substantial harmful impacts on crop output [5]. The deterioration of soil structure and permeability, which impede water passage to the root zone and disrupt seedling emergence, delaying plant development in sodic-alkaline soils, has a particular impact on agricultural production [6,7]. These soils contain significant levels of alkaline salts (NaHCO_3_ and Na_2_CO_3_), which elevate the pH of the soil solution through alkaline hydrolysis [8]. These alkaline salts can cause more damage and serious severe effects on plants than those occurring by neutral salts (NaCl and Na_2_SO_4_) [9,10]. Alkali stress can intensely increase oxidative damage by the generation of huge amounts of reactive oxygen species (ROS), leading to destroying the nucleic acids, proteins, and lipids [11,12]. Alkaline environments also influence the availability of various macro and micro nutrients that plants require [13]. Furthermore, sodic-alkaline stress can seriously affect a broad spectrum of essential physiological and molecular aspects of plant growth and development [12,14,15]. Affecting photosynthesis, plant water relation, respiratory metabolism, phytohormones homeostasis, organic acids metabolism, and expression of Na detoxification genes are common responses to sodic-alkaline stress in various plant species [12,15,16]. Plants adapted to these environments possess multiple physiological, biochemical and molecular aspects to tolerate ionic and drought stress [12,17]. Generally, these features can be reinforced using some exogenous compounds which act as efficient antioxidants or signaling molecules under different abiotic stresses [18,19,20].

Alpha-lipoic acid (1,2-dithiolane-3-pentanoic acid; ALA) is found in both bacterial and eukaryotic organisms [21]. Because of its antioxidant and therapeutic characteristics, it has attracted a lot of attention as a dietary supplement for humans [22,23]. Its two sulfhydryl moieties, which enable it to scavenge free radicals and bind metals, are responsible for its antioxidant capacity [24]. Exogenous ALA can improve photosynthetic performance, mitigate lipid peroxidation, stimulate the antioxidant systems and regulate the osmolytes under several abiotic stresses [25,26,27,28]. Moreover, ALA can decrease the cytotoxic effects of Na and maintain the ionic homeostasis in the salt-stressed sorghum plants; these influences were associated with altering the expression of ionic transport-related genes, i.e., plasma membrane Na^+^/H^+^ antiporters protein (*SOS1*), vacuolar Na^+^/H^+^ antiporters protein (*NHX1*) and the high-affinity potassium transporter protein (*HKT1*) [28]. Under osmotic stress, exogenous ALA increased the activities of enzymatic antioxidants such as superoxide dismutase (SOD), catalase (CAT), guaiacol peroxidase (GPX), glutathione reductase (GR), and monodehydroascorbate reductase (MDHAR) [21,26,29]. Additionally, ALA can be involved in the improvement of yield and restoration of grain quality attributes of drought-stressed wheat plants [25].

The effect of ALA on wheat plants is still unknown under sodic alkaline stress. Therefore, this study aimed to understand the possible and protective roles of ALA as a promising and potent dithiol antioxidant on growth, water status, oxidative injuries and; several biochemical and molecular of alkali-stress relevant aspects in wheat plants. These findings can present evidence that ALA is involved in plant tolerance to sodic alkaline stress.

## 2. Materials and Methods

### 2.1. Plant Material and Growth Conditions

Wheat grains (*Triticum aestivum* L., Cv. Giza 168) produced by the Agriculture Research Center, Egypt, were sterilized using a solution of 0.1% NaOCl for 5 min, then rinsed 4 times with distilled water. Wheat grains were sown in plastic pots (20 cm diameter × 25 cm height). The pots were filled with 10 kg pre-washed sand. All pots were kept under greenhouse conditions; the average air temperature (T_ave_) was 24.8 °C, where the T_max_ during the day was 30.4 and T_min_ at night was 19.2. Meanwhile, the relative humidity was 70.5 ± 2 (Data were recorded by a digital Thermo/hygrometer Art placed in the middle of the greenhouse; No.30.5000/30.5002, TFA; Germany). All pots were regularly irrigated with half-strength Hoagland’s nutrient solution. The irrigation was implemented three times a week after calculating the reduction in the water holding capacity (WHC) by weight method.

### 2.2. Treatments and Experimental Design

After three weeks from sowing, sodic alkaline stress was applied as described by Han et al. [16], where the alkali-stressed seedlings were irrigated by a modified half-strength Hoagland nutrient solution containing two alkaline salts (NaHCO_3_ and Na_2_CO_3_) that were mixed in a ratio of 1:1 (50 mM; pH 9.7); while the non-stressed seedlings were irrigated with a half-strength Hoagland’s nutrient solution. To investigate the effect of α-lipoic acid (ALA) on wheat seedlings grown under sodic alkaline stress, the two groups of alkali-stressed and non-stressed seedlings were divided into two subgroups. The first subgroup was sprayed with distilled water (ALA-untreated seedlings), while the second subgroup was sprayed with 20 µM ALA (Cayman Chemical Company, Ann Arbor, MI, USA). Tween-20 was used at 0.05% as a surfactant agent for all foliar treatments. Each pot containing 5 seedlings was sprayed for 7 consecutive days at 5.30 pm with 30 mL of the spray solution. Seedlings were allowed to grow for another week; then, samples were gathered to determine the growth and different studied parameters. The experiment was included 4 treatments: (i) The control, (ii) sodic alkaline stress, (iii) ALA and (iv) sodic alkaline stress + ALA. The experimental layout was a completely randomized design (CRD) with three replicates. The total number of pots was 72 (4 treatments × 6 pots × 3 replicates).

### 2.3. Plant Growth and Leaf Pigments Composition

At 35 days after sowing, shoot fresh weight was determined using a digital balance, while shoot dry weight was recorded after full drying at 105 °C in an air-forced vented oven. The concentration of chlorophyll (Chl) a and b, and carotenoids in fresh leaves was determined spectrophotometrically according to Lichtenthaler and Wellburn [30]. Fresh weight (0.05 g) of fully expanded young leaves was used for pigment extraction in 80% acetone. The extract of pigments was measured versus a blank of pure 80% acetone at 663, 644, and 452.5 nm for Chla, Chlb, and carotenoid contents, respectively.

### 2.4. Leaf Relative Water Content (RWC) and Osmolytes

Leaf relative water content (RWC) was estimated according to Abd El-Gawad et al. [31]. Leaf discs from 6 of the fully expanded leaves were weighed (FW) and placed immediately in distilled water for 2 h at 25 °C, then the turgid weight (TW) was recorded. After that, discs were fully dried in an oven at 110 °C for 24 h (DW). Relative water content (RWC) was calculated using the following formula:$$\mathrm{RWC}\;\left(\%\right)\;=\;\frac{\mathrm{FW}-\mathrm{DW}}{\mathrm{TW}-\mathrm{DW}}\times 100$$

Proline was spectrophotometrically estimated at 520 nm using the ninhydrin assay according to Bates et al. [32]. Total soluble sugars were quantified at 490 nm by phenol and sulphuric acid method as described by Chow and Landhäusser [33].

### 2.5. Lipid Peroxidation and Hydrogen Peroxide

Lipid peroxidation was quantified by measuring the peroxidation products, thiobarbituric acid reactive substances (TBARS). TBARS was quantified by the method of Abd Elbar et al. [34] by reading the developed color at 535 nm and 600 nm. Hydrogen peroxide was quantified by the colorimetric method of potassium iodide as described by Velikova et al. [35]. A fresh weight (0.5 g) of leaf tissues was homogenized in 3 mL of 1% (*w*/*v*) trichloroacetic acid (TCA). The homogenate was centrifuged at 10,000 rpm at 4 °C for 15 min. Subsequently, 0.75 mL of the supernatant was added to 0.75 mL of 10 mM potassium phosphate buffer (pH 7.0) and 1.5 mL of 1 M KI. The mixture was measured at 390 nm.

### 2.6. Total Soluble Phenols and Ascorbic Acid

Total soluble phenols were determined using the method of Folin-Denis reagent by reading the developed blue color at 725 nm as described by Skalindi and Naczk [36]. Ascorbic acid was determined using the titrimetric method of 2, 6-Dichloroindophenol [37].

### 2.7. Total Soluble Protein and Enzyme Assays

Fresh leaves (0.2 g) were grounded in 4 mL of 0.1 M ice-cold sodium phosphate buffer (pH 7.0) containing 1% (*w*:*v*) polyvinylpyrrolidon (PVP) and 0.1 mM EDTA, centrifuged at 10,000× *g* at 4 °C for 20 min. The supernatant was used for the next enzyme activity assays. To calculate the specific activity of different enzymes, the total soluble protein was also determined in the supernatant, according to Bradford [38]. Superoxide dismutase (SOD; EC 1.15.1.1) activity was evaluated according to the ability to inhibit the photochemical reduction of nitro blue tetrazolium (NBT) at 560 nm [39]. Catalase (CAT; EC 1.11.1.6) activity was assayed by monitoring the decrease in absorbance of H_2_O_2_ at 240 nm [40]. Guaiacol peroxidase (G-POX; EC1.11.1.7) activity was evaluated by observing its ability to convert guaiacol to tetraguaiacol by monitoring the increase in absorbance at 470 nm [41]. Ascorbate peroxidase (APX; EC 1.11.1.11) activity was determined based on the decrease in ascorbate at 290 nm [42].

### 2.8. Determination of Na, K and Ca

The flame photometric method (Jenway, UK) was used to determine the nutrients (Na, K and Ca) concentrations [43].

### 2.9. Gene Expression by qRT-PCR

Total mRNA was extracted from different treatments using 0.5 g fresh leaves and a total RNA extraction kit (Sigma-Aldrich, Darmstadt, Germany) according to the manufacturer’s protocol. The purified RNA was quantitated spectrophotometrically and checked on 1% agarose gel. Reverse transcription of RNA was performed. The reaction mixture contained 10 as oligo dT primer (10 pml/μL), 2.5 μL 5X buffer, 2.5 μL MgCl_2_, 2.5 μL 2.5 mM dNTPs, 4 μL from oligo (dT), 0.2 μL (5 Unit/μL) reverse transcriptase (Promega, Walldorf, Germany) and 2.5 μL RNA. RT-PCR amplification was performed in a thermal cycler PCR, programmed at 42 °C for 1 h and 72 °C for 20 min. For each sample, total RNA (5 μg) was reverse transcribed to complementary cDNA in a reaction mixture consisting of 2.5 μL 2.5 mM dNTPs, 2.5 μL MgCl_2_, 1.0 μL oligo dT primer (10 pml/μL), 2.5 μL 5X buffer, 0.2 μL (5 Unit/μL) reverse transcriptase (Promega, Walldorf, Germany), RT-PCR amplification was performed in a thermal cycler PCR, at 42 °C for 1.5 h and 80 °C for 20 min. Quantitative real-time PCR was carried out on 1 μL diluted cDNA by triplicate using the real-time analysis using (Rotor-Gene 6000, Germany) system and the primer sequences used in qRT-PCR were given in Table 1. Primers of Δ pyrroline-5-carboxylate synthase (P5CS), photosystem II D2 protein (PsbD), the plasma membrane Na^+^/H^+^ antiporters protein of salt overly sensitive gene (SOS1), tonoplast-localized Na^+^/H^+^ antiporter (NHX1) protein and β-Actin housekeeping gene (reference gene) were used for gene expression analysis using an SYBR^®^ Green-based method. A total reaction volume of 20 µL was used. The reaction mixture consists of 2 µL of template, 10 µL of SYBR Green Master Mix, 2 µL of reverse primer, 2 µL of forward primer, and sterile dist. water for a total volume of 20 µL. PCR assays were performed using the following conditions: 95 °C for 15 min followed by 40 cycles of 95 °C for 30 s and 60 °C for 30 s. The CT of each sample was used to calculate ΔCT values (target gene CT subtracted from β-Actin gene CT). The relative gene expression was determined using the 2-ΔΔCt method [44].

### 2.10. Statistics

One way ANOVA procedure was followed using SAS [45] software. Means ± SD were calculated from three replicates and Duncan’s multiple range test (*p* ≤ 0.05) was used to determine significant differences between means.

## 3. Results

### 3.1. Effect of ALA on Plant Growth and Chlorophyll Composition under Sodic Alkaline Stress

Wheat seedlings exposed to sodic alkaline stress obviously exhibited a significant (*p* ≤ 0.05) inhibition in plant growth (shoot fresh and dry weight) compared to the unstressed seedlings (Figure 1A,B). Similarly, this effect was observed in Chl a, Chl a + b and Chl a/b ratio, while Chl b was more stable under sodic alkaline stress (Figure 1C–F). Under unstressed conditions, applied-ALA led to greater and significant (*p* ≤ 0.05) improvement in growth parameters, Chl a and Chl a + b. However, this enhancement was insignificant (*p* ≤ 0.05) in the Chl b and Chl a/b ratio. Meanwhile, applied-ALA under sodic alkaline stress conditions led to a significant (*p* ≤ 0.05) increase in growth parameters, Chl a and Chl a/b ratio compared to the ALA-untreated seedlings.

### 3.2. Effect of ALA on Leaf Relative Water Content (RWC) and Osmolytes under Sodic Alkaline Stress

Sodic alkaline stress significantly (*p* ≤ 0.05) reduced RWC compared to the unstressed conditions (Figure 2). An opposite and significant (*p* ≤ 0.05) trend was observed with respect to proline and total soluble sugars. Wheat seedlings treated by ALA significantly (*p* ≤ 0.05) demonstrated a considerable increase in RWC and total soluble sugars compared to the ALA-untreated seedlings under sodic alkaline stress conditions. However, proline was significantly decreased (*p* ≤ 0.05) in this respect.

### 3.3. Effect of ALA on the Oxidative Damage Induced by the Sodic Alkaline Stress

Compared with the control, wheat seedlings exposed to sodic alkaline stress revealed a higher and more significant (*p* ≤ 0.05) rate of lipid peroxidation and extensive accumulation of reactive oxygen species (ROS) as indicated by the increase in the concentration of malondialdehyde (MDA) and H_2_O_2,_ respectively (Figure 3). However, applied-ALA significantly (*p* ≤ 0.05) decreased MDA and H_2_O_2_ compared to the ALA-untreated plants.

### 3.4. Effect of ALA on the Non-Enzymatic Antioxidants under Sodic Alkaline Stress

Except for carotenoids, all studied non-enzymatic antioxidants, including carotenoids, total soluble phenols, and ascorbic acid, were significantly (*p* ≤ 0.05) increased by exposure to sodic alkaline stress (Figure 4). Under sodic alkaline stress conditions, applied-ALA led to improve the carotenoids, total soluble phenols and ascorbic acid. A similar trend was observed under non-stressed conditions.

### 3.5. Effect of ALA on Antioxidant Enzymes under Sodic Alkaline Stress

Sodic alkaline stress significantly (*p* ≤ 0.05) enhanced the activities of antioxidant enzymes (SOD, CAT, G-POX and APX) compared to the unstressed conditions (Figure 5). Wheat seedlings treated by ALA exhibited an obvious improvement in SOD, G-POX and APX, but CAT was decreased under non-sodic alkaline conditions. On the other hand, the general tendency under sodic alkaline conditions was that SOD and APX were improved in ALA-treated seedlings, while the activities of CAT and G-POX were inhibited compared to ALA-untreated seedlings.

### 3.6. Effect of ALA on Leaf Na, K, Na/K Ratio and Ca Concentration under Sodic Alkaline Stress

The leaf concentration of Na and Na/K ratio was interestingly and significantly (*p* ≤ 0.05) increased in the seedlings under sodic alkaline conditions compared to the unstressed ones, whereas K and Ca were dramatically decreased (Figure 6). Wheat seedlings of ALA treatment showed a significant (*p* ≤ 0.05) decrease in Na and Na/K ratio compared to ALA-untreated seedlings under sodic alkaline conditions. In contrast, K and Ca were significantly improved in this respect. Additionally, it was observed that the concentration of Ca was decreased by the treatment of ALA compared to the ALA-untreated plants under non-stressed conditions.

### 3.7. Effect of ALA on the Relative Expression of Genes under Sodic Alkaline Stress

To understand the protective role of ALA in the improvement of plant tolerance to sodic alkaline stress, the relative gene expression of *D2-protein*, *P5CS*, *SOS1* and *NHX1* was studied by RT-qPCR (Figure 7). It can be observed that sodic alkaline stress significantly (*p* ≤ 0.05) down-regulated the relative expression of *D2-protein*, while *P5CS*, *SOS1* and *NHX1* were up-regulated. Interestingly, wheat seedlings of ALA treatment showed significant up-regulation of *D2-protein* in both non-stressed and sodic alkaline stressed seedlings. Meanwhile, this response was in parallel with a significant down-regulation of *P5CS*, *SOS1* and *NHX1* under sodic alkaline stress conditions.

## 4. Discussion

Exposing plants to sodic alkaline salts (Na_2_CO_3_ and/or NaHCO_3_) is more mischievous than neutral salts (NaCl and/or Na_2_SO_4_) due to the presence of high pH of alkaline stress in addition to the toxic effect of Na ions that can hinder plant growth and development [12,15,17]. In this study, we found that wheat seedlings exposed to sodic alkaline stress obviously exhibited a significant (*p* ≤ 0.05) decrease in plant growth (shoot fresh and dry weight) compared to the unstressed seedlings (Figure 1A,B). This effect could be attributed to the disturbance of nutrient uptake and the massive damage of the root system in the alkaline soil [17,46,47]. As well as Chl a, Chl a + b and Chl a/b ratio were substantially decreased; while Chl b was more stable under sodic alkaline stress (Figure 1C–F). Affecting the chlorophyll content and reducing of Chl a/b ratio under various stress conditions are common biochemical signs to induce the oxidative damage and destruction of photosynthetic machinery [25,28,48,49]. Due to its essential role as a major electron donor in the electron transport chain, reducing Chl a can be considered an important regulatory step under stress conditions. This response can avoid the over reduction in the photosynthetic electron transport chain and consequently restrict the excessive generation of ROS [50]. Meanwhile, Chl b plays a protective role for the photosynthetic apparatus under stress conditions [51,52].

Exogenous ALA can enhance plant growth under osmotic stress conditions, i.e., drought and salinity [25,28]. This effect could be attributed to its high efficiency as a strong soluble antioxidant in both hydrophilic and hydrophobic phases [53]. Moreover, it is very effective in both its forms of reduced dihydrolipoic acid (DHLA) and oxidized α-lipoic acid (ALA) [27,53]. ALA can repair photosynthesis and stimulate the biosynthesis of photosynthetic pigments under stress conditions [25,27,28]. In a previous study on salt-stressed canola seedlings, exogenous ALA was found to promote the root system [54]. This effect can help the shoot system to grow by increasing the water and nutrients uptake.

In the current study, enhancing the leaf relative water content (RWC) under sodic alkaline stress was observed in ALA-treated seedlings compared to the ALA-untreated ones (Figure 2A). This improvement was in parallel with a greater accumulation of soluble sugars of sodic alkaline stressed seedlings. Meanwhile, wheat alkaline stressed seedlings treated by ALA demonstrated an obvious decrease in proline compared to ALA-untreated ones (Figure 2B,C). It is well documented that under osmotic stress conditions, plants accumulate a high concentration of proline, soluble sugars, amino acids and other organic solutes [31,55]. This mechanism enables plants to regulate the osmotic potential and enhance the water uptake under stress conditions [21,56]. In this study, the accumulation of proline and soluble sugars is differently regulated by ALA treatment in the sodic alkaline stressed seedlings. This response indicates that ALA might strengthen the pathway of carbon metabolism in a different way from nitrogen under sodic alkaline stress.

The application of exogenous ALA led to a significant (*p* ≤ 0.05) decrease in MDA and H_2_O_2_ under the sodic alkaline stress conditions, while no change was observed under non-alkaline conditions (Figure 3A,B). ALA possess two sulfhydryl moieties with an enormous antioxidant capacity [21]. It has been found that ALA can mitigate the oxidative damage induced by various abiotic stresses, i.e., drought [25,27], salinity [28,54], osmotic stress [21] and heavy metals [57]. However, no significant (*p* ≤ 0.05) change was observed in MDA and H_2_O_2_ of ALA-treated and non-treated seedlings under non-alkaline conditions. It has been suggested that MDA and ROS can trigger damage or protection depending on their cellular concentration. Among the benefits of MDA and ROS under non-stressed conditions, their indispensable role in regulating the signaling and the other biological functions such as plant growth and development, the mitotic activity in meristems and the flower opening [58,59].

In this study, we found an obvious decrease in the carotenoid content under sodic alkaline stress (Figure 4A). This shortfall might be due to the increase in ABA hormone, which is synthesized from the same precursors of 9′-cis-neoxanthin and 9-cis-violaxanthin under osmotic stress conditions [60]. In this context, ABA has been shown to enhance the tolerance to alkaline stress in rice seedlings [61]. Therefore, enhancing the carotenoid content in the ALA-treated seedlings could indicate its ability to enhance the tolerance to sodic alkaline stress and consequently restrict the excessive biosynthesis of ABA under stressed conditions. In contrast, the increase in total soluble phenols under sodic alkaline stress (Figure 4B) was earlier evidenced as protective molecules under various stress conditions, including drought [62,63], salinity [64] and alkaline stress [17]. This effect is due to their properties as efficient antioxidants, as well as their function as a source for carbon supplies under stress conditions [63]. The improvement of total soluble phenols in the ALA-treated seedlings either under alkaline or non-stressed conditions may indicate the positive effect of ALA on photosynthesis, which is responsible for the presence of carbon skeleton (shikimic acid pathway) of all secondary metabolites [65]. The content of ascorbic acid was also enhanced under sodic alkaline stress (Figure 4C). It has been reported that ascorbic acid biosynthesis-related enzymes such as DHAR, MDHAR GMP, GME and GalUR were upregulated under stress conditions [66]. This effect may maintain ROS under control and enhance the ascorbate–glutathione cycle [67,68]. Increasing the concentration of ascorbic acid with an adequate content of dihydrolipoic acid (DHLA) can maintain the redox homeostasis of barley plants under salinity stress in the presence of high activity of ascorbate peroxidase (APX) [29].

In addition to the non-enzymatic antioxidants, plants develop several antioxidative enzymatic systems to overcome the oxidative damage induced by environmental challenges. In this study, we found that sodic alkaline stress increased the activities of SOD, CAT, G-POX and APX compared to the unstressed conditions (Figure 5A–D). These responses can help plants to survive under stress conditions by scavenging ROS and maintaining them under control [18,28,49,69]. However, alkaline stressed seedlings treated by ALA demonstrated greater activity in SOD and APX than the ALA-untreated seedlings. Meanwhile, a significant decrease was observed in CAT and G-POX of ALA-treated seedlings under alkaline stress conditions. These effects imply that under sodic alkaline stress, ALA can reduce the cytotoxicity of superoxide anion and H_2_O_2_ by affecting the activities of SOD and APX, respectively [68,69,70,71,72,73,74]. Several previous studies have confirmed that exogenous ALA can alter the activities of antioxidant enzymes in different patterns according to its concentration, plant species and the type of environmental stress [25,26,28]. In the current study, we found a close link between exogenous ALA and reducing the oxidative damage (Figure 3A,B) on one side and improving the antioxidative capacity (non-enzymatic/ enzymatic antioxidants) of sodic alkaline stressed seedlings on the other side (Figure 4 and Figure 5).

On the other hand, wheat seedlings exposed to sodic alkaline stress demonstrated a higher accumulation of Na with a serious decrease in K and Ca, leading to a considerable increase in the Na/K ratio (Figure 6A–D). This disturbance in the ionic homeostasis, in particular with elevating the concentration of Na, can be toxic to the cytosolic enzymes [70]. Moreover, it can affect plant water relations and hinder the uptake of nutrients [28,49,55]. Additionally, the higher values of pH under alkaline stress can impede or prevent the uptake of essential nutrients [17]. In this study, we noticed that exogenous ALA achieved an improvement in the leaf water status as indicated by increased RWC and osmolytes (Figure 2A–C). Furthermore, ALA reduced the oxidative damage in leaf tissues by enhancing the antioxidative capacity of plants (Figure 4 and Figure 5). These effects can positively affect the selective permeability of the cell membranes and consequently the uptake of water and the essential nutrients such as K and Ca with the transpiration stream. In this context, similar results have been observed on different plant species [21,26,28,54].

To further understand the role of ALA in wheat plant tolerance to sodic alkaline stress, molecular studies using qRT-PCR were carried out to explore the relative expression of four genes with different functions: D2-protein, which is considered one of the major core proteins in the photosystem II center reaction [71], Δ^1^-pyrroline-5-carboxylate synthetase (*P5CS*), which is responsible for the synthesis of proline from glutamate under stress conditions [75], the plasma membrane Na^+^/H^+^ antiporters protein (*SOS1*) [76] and the vacuolar Na^+^/H^+^ antiporters protein (*NHX1*) [76]. Our findings indicated that sodic alkaline stress down-regulated D2-protein. Meanwhile, *P5CS*, *SOS1* and *NHX1* were down-regulated. These effects could be attributed to oxidative damage, which has been evidenced earlier in this study (Figure 2A–C). Since reducing photosynthesis and an increasing accumulation of proline and Na are common markers under saline and alkaline stress conditions in several plant species [17,18,28,55]. In contrast, applied-ALA significantly reversed these responses specifically under sodic alkaline stress. These findings may elucidate the protective role of ALA against sodic alkaline stress in wheat plants.

## 5. Conclusions

Generally, the results of this study may be considered the first conclusive evidence of the protective roles of ALA as a promising and potent antioxidant in mitigating the deleterious effects of sodic alkaline stress in wheat plants. These positive effects included the improvement of growth, water uptake, ionic homeostasis and photosynthesis. Moreover, ALA reduced the alkalinity induced oxidative damages through activating a wide spectrum of antioxidative defense systems (enzymatic and non-enzymatic). These responses were also associated with altering the expression of tolerance and sodic alkaline stress-responsive genes (Figure 8). Further studies using advanced molecular techniques are required to better understand the role of ALA in inducing the signaling pathways that improve plant tolerance to sodic alkaline stress in plants.

## Figures and Tables

**Figure 1 plants-11-00787-f001:**
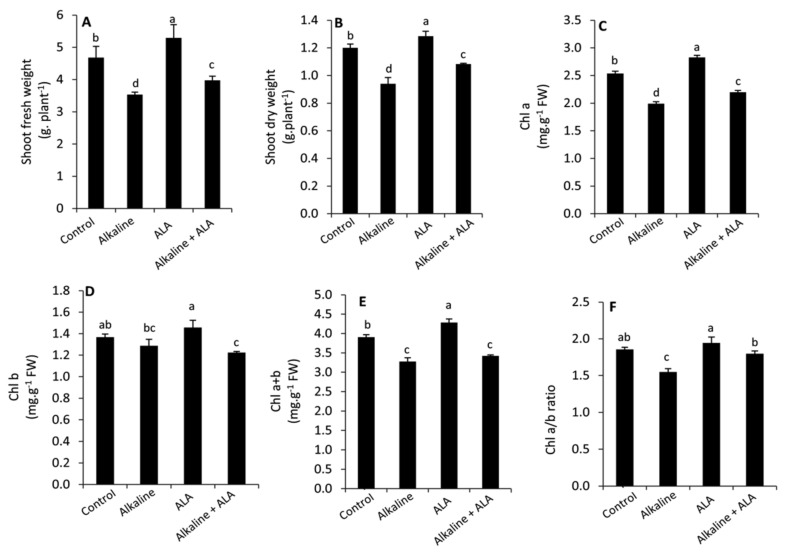
Effect of exogenous α-lipoic acid (ALA; 20 µM) on shoot fresh weight (**A**), shoot dry weight (**B**), Chl a (**C**), Chl b (**D**), Chl a + b (**E**) and Chl a/b ratio (**F**) of wheat seedlings grown under sodic alkaline stress. Chl, chlorophyll.

**Figure 2 plants-11-00787-f002:**
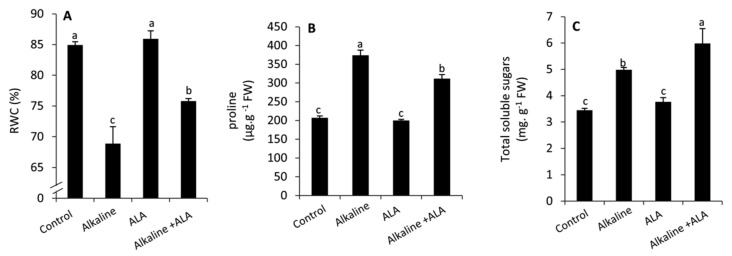
Effect of exogenous α-lipoic acid (ALA; 20 µM) on RWC (**A**), proline (**B**), total soluble sugars (**C**) of wheat seedlings grown under sodic alkaline stress. RWC, leaf relative water content.

**Figure 3 plants-11-00787-f003:**
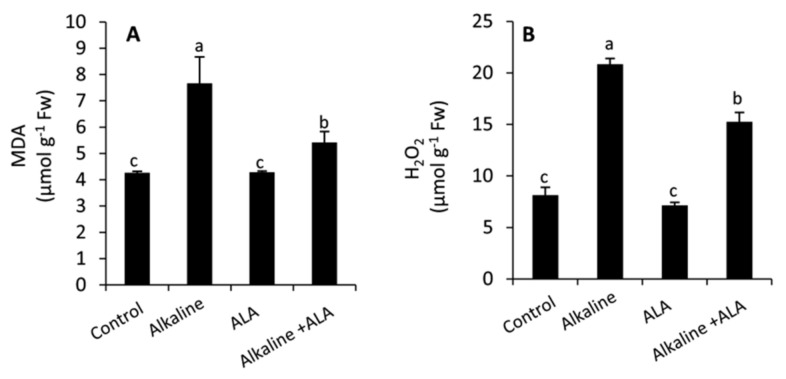
Effect of exogenous α-lipoic acid (ALA; 20 µM) on the concentration of malondialdehyde; MDA (**A**) and H_2_O_2_ (**B**) in the leaves of wheat seedlings grown under sodic alkaline stress.

**Figure 4 plants-11-00787-f004:**
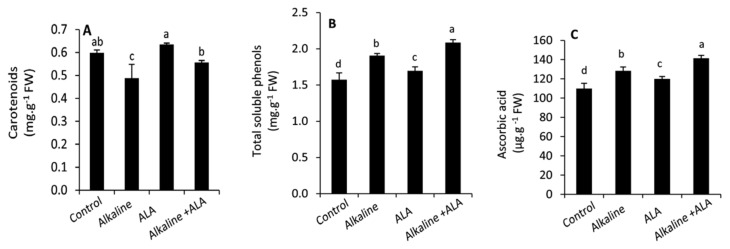
Effect of exogenous α-lipoic acid (ALA; 20 µM) on the concentration of carotenoids (**A**) total soluble phenols (**B**) and ascorbic acid (**C**) in the leaves of wheat seedlings grown under sodic alkaline stress.

**Figure 5 plants-11-00787-f005:**
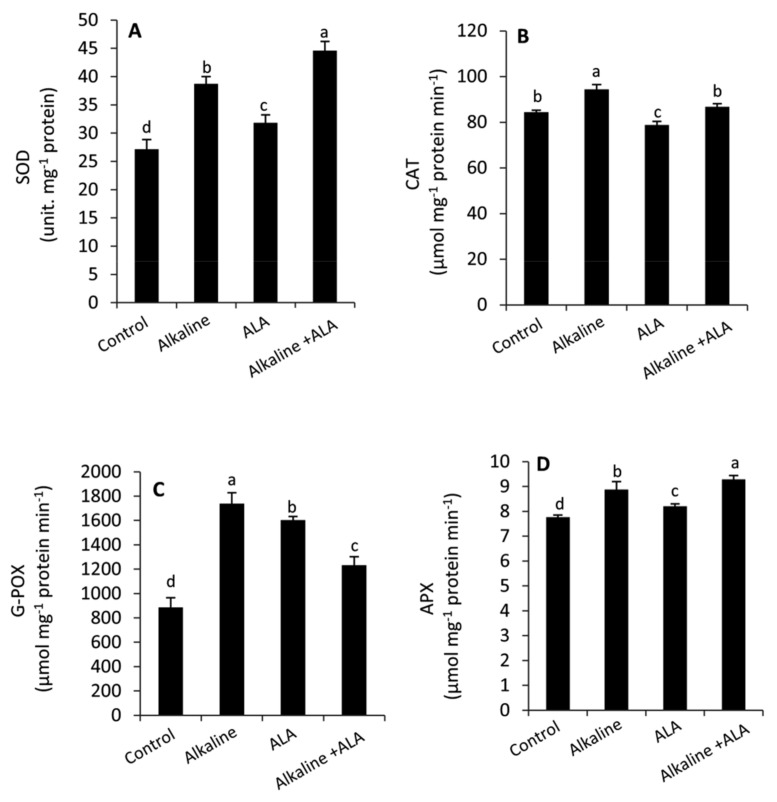
Effect of exogenous α-lipoic acid (ALA; 20 µM) on the activities of antioxidant enzymes: Superoxide dismutase; SOD (**A**) Catalase; CAT (**B**) Guaiacol peroxidase; G-POX (**C**) and Ascorbate peroxidase; APX (**D**) in the leaves of wheat seedlings grown under sodic alkaline stress.

**Figure 6 plants-11-00787-f006:**
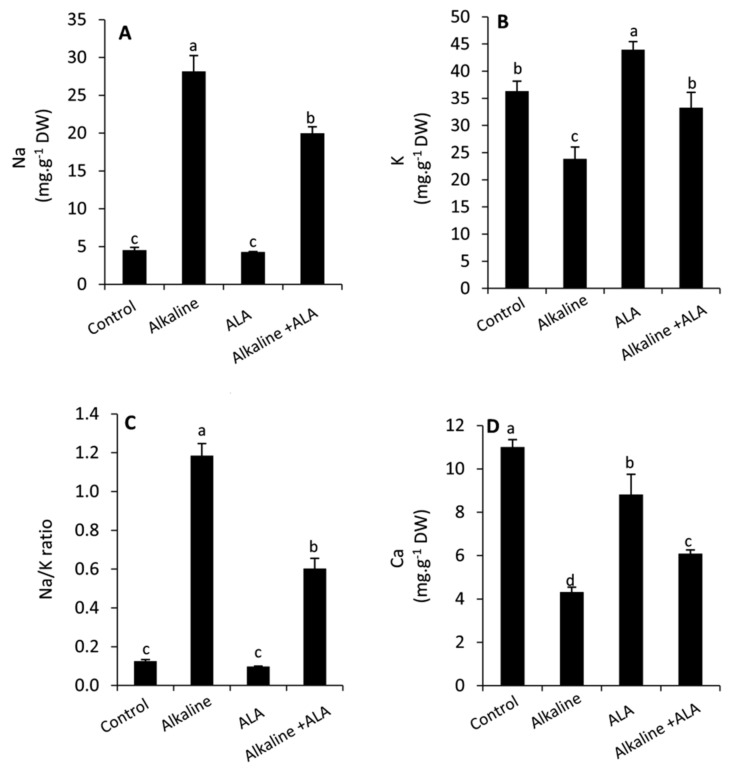
Effect of exogenous α-lipoic acid (ALA; 20 µM) on the concentration of Na (**A**) K (**B**) Na/K ratio (**C**) and Ca (**D**) in the leaves of wheat seedlings grown under sodic alkaline stress.

**Figure 7 plants-11-00787-f007:**
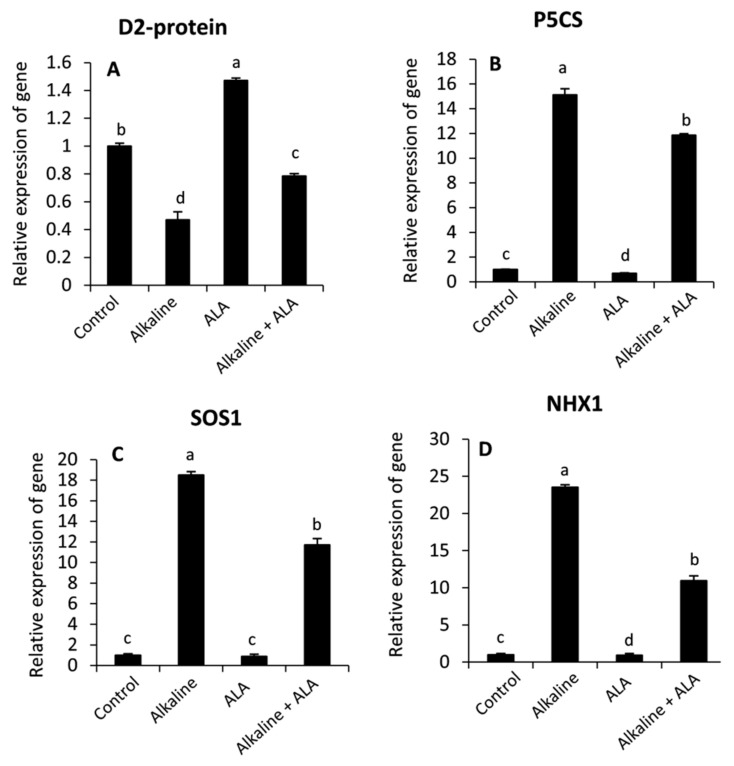
Effect of exogenous α-lipoic acid (ALA; 20 µM) on the relative expression of photosystem II D2 protein; PsbD (**A**), Pyrroline-5-carboxylate synthase; P5CS (**B**), plasma membrane Na^+^/H^+^ antiporters protein of salt overly sensitive gene; SOS1 (**C**) and tonoplast-localized Na^+^/H^+^ antiporter protein; NHX1 (**D**) using quantitative real time PCR in the leaves of wheat seedlings grown under sodic alkaline stress.

**Figure 8 plants-11-00787-f008:**
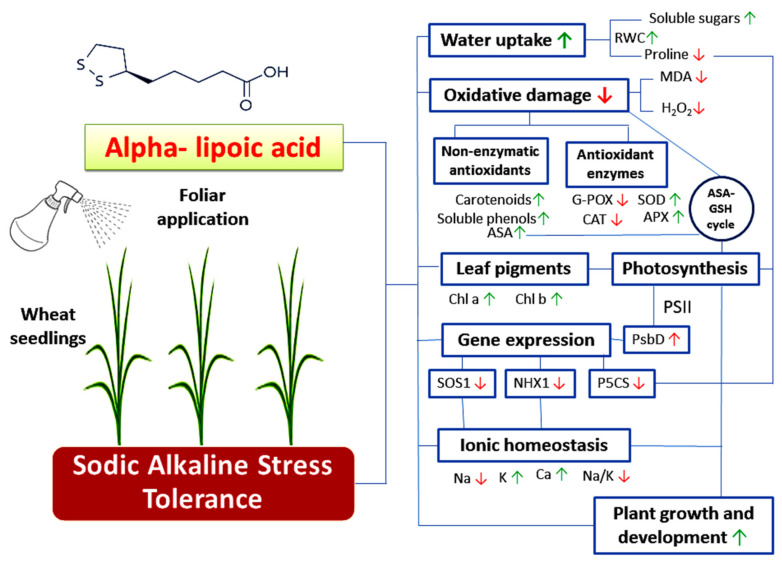
A summary for the induction of sodic alkaline stress tolerance in wheat seedlings by exogenous application of α-lipoic acid. Arrows up and down refer to upregulation and downregulation of various studied aspects respectively.

**Table 1 plants-11-00787-t001:** Oligonucleotides primer pairs used for quantitative RT-PCR analysis.

Gene Name		Sequence	NCBI Accession
D2-protein (PsbD)	F	5′-CGCTTTAGGGGGTGTGTTTA-3′	NC_002762.1
	R	5′-GCCCCCATAGTAGCAACAAA-3′	
P5CS	F	5′-TCGGTGCTGAGGTTGGCATAAG-3′	JQ063079.1
	R	5′-TTGTCACCATTCACCACTTGCCC-3′	
SOS1	F	5′-GTTGTCGGTGAGGTCGGAGGG-3′	AY326952
	R	5′-TCATCTTCTCCTACCGCCCTGC-3′	
NHX1	F	5′-CACCAGCCACGGATCTTTCT-3′	AY461512.1
	R	5′-TTCACGATCAGTGGAGTGCC-3′	
Actin	F	5′-TGCTATCCTTCGTTTGGACCTT-3′	AB181991
	R	5′-AGCGGTTGTTGTGAGGGAGT-3′	

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
