# Peer review of "Alpha Lipoic Acid as a Protective Mediator for Regulating the Defensive Responses of Wheat Plants against Sodic Alkaline Stress: Physiological, Biochemical and Molecular Aspects"

_plants, 2022, doi:10.3390/plants11060787_

Round 1

Reviewer 1 Report

General Comments:

The research article is designed to study the “Alpha Lipoic acid as a Protective Mediator for Regulating the Defensive Responses of Wheat Plants against Sodic Alkaline Stress: Physiological, Biochemical and Molecular Aspects”. The author’s mainly used α-Lipoic acid to improve the tolerance of wheat crop. The discrepancies of the article are mentioned below to improve its quality.

Revisions:

The authors must the problem statement of this particular study and also explain the objective of the article.

L32, unblood the bold sentence.

In the abstract, the methodology of the study is not mentioned so kindly mention of methodology of the study in 2-3 lines. The abstract should be cover all aspects of the study. Intro, objective, methodology, results and discussion. Moreover, at least provide one core statistical results of your study in the abstract. Kindly revise your abstract.

L46-47, In keywords, don’t repeat those words which are already used in the title of the article, i.e., Wheat; α-lipoic acid; Alkali stress.

L53-55, rewrite these line. You have mentioned the repeated information.

L63-64, mention the full name of the alkaline salts, i.e., NaHCO3 and Na2CO3 once in your article where they have used earlier.

L67, use “micro nutrients” instead of “micronutrients”.

The introduction section is needs more improvement and more details about the literature. In the last paragraph of the article, clearly mention the objective of your study and novelty statement.  

L96, washed several times? Kindly mention the quantity, i.e., 20 or 30 times

L98, remove the “of” from line.

I would recommend splitting the heading of “2.1. Growth Conditions and Experimental Design” into two separate headings. Explain the details of the experiment in a much-arranged manner under separate headings, 2.1 Growth Conditions and 2.2 Experiment Design.

L120, The total number of pots was 72. Check the grammar of this sentence and throughout the manuscript.

L123, The shoot's fresh weight was measured after how many growing days? Add information.

Check the superscript and subscript throughout the article, i.e., L193, 950 C and L179 72°C. Use the correct annotations for superscript and subscript.

L222. You have mentioned the statement i.e., “Bars represent standard deviation (SD) of the means (n = 3). Different letters indicate significant differences among the treatments at P ≤ 0.05, according to Duncan’s multiple range test.” This line is mentioned same as it is at the end of every figure caption. There is no need of this statement after every figure. You have mentioned it in the methodology, which is fine. You can more explain it in methodology but don’t repeat the same line in the caption of every figure.

Statistical results are missing in the article. The data you have collected and its effects must be proven with statistical analysis.

Rewrite the L433-435 for a clear understanding of the readers.

Author Response

Dear reviewer I

Thank you so much for your valuable comments which helped us to improve the overall quality of our manuscript. All your comments and suggestion have taken into account in the revised version of our manuscript

please find the attached file of our cover letter

Regards

the authors

Reviewer 2 Report

Dear Editor/Authors,

In the present manuscript, the authors studied the effects of Alpha Lipoic acid as a potential defense in  wheat cultivar cv. Giza in sodic-alkaline stress. The manuscript is an interesting preliminary research article. For a better understanding and improvement of this article, we will suggest addressing the following issues.

In the introduction section, please list the aims of the work in detail. 

On what basis was this cultivar selected for the research?Was the wheat selected for the study previously analyzed for stress tolerance? Do you plan to expand the number of analyzed varieties?

Please edit the conclusions so that they contain the most important conclusions of the work- "highlighter the novelty of manuscript" and not just a summary of the results (what has decreased and what has increased).

Author Response

Dear reviewer II

We would like to thank you so much for your valuable comments which helped us to improve the overall quality of our manuscript 

Please find the attached file of our responses on all comments and suggestions point by point

Regards

the authors

Round 2

Reviewer 1 Report

No change required.